# SlideGen: A Multi-Agent Framework for Automatic Scientific Slide Generation

## Abstract

Generating academic slides from scientific papers is often challenging as it requires reasoning over long context and carefully planning layouts. However, most prior work just treat it as a text summarization task, overlooking the inherent complexity of visual design. To tackle this challenge, we propose **SlideGen**, a modular, visual-in-the-loop agentic pipeline for paper-to-slide generation, which utilizes six VLM workers to collaborate together. It plans the outline (Outliner), matchs figures/tables/equations to outline bullets (Mapper/Formulizer), lays out pages via template selection (Arranger), writes notes (Speaker), and refines with merging and emphasis (Refiner). To better evaluate the quality of the generated slides, we further release the **Paper2Slide Benchmark** of paper–slide pairs and provide automated evaluation protocols: *(i)* Visual Aesthetics – a geometry-aware density score for layout balance and spacing, *(ii)* Holistic Assessment – a VLM-as-judge criteria over content, design, and coherence, enabling reliable, end-to-end assessment; and *(iii)* Communication Effectiveness – we use SlideQA, a question answering task that measures the ability of presentation slides to convey information; *(iv)* Textual Coherence – textual fluency. Across a diverse set of strong baselines, **SlideGen** demonstrates strong results across all evaluation metrics and outperforms various competing methods, offering human-level slide-making capabilities. Our framework identifies promising directions for building the next generation of end-to-end slide generators. The code is available for full reproducibility at Anonymous Github.

## 1 Introduction

Slide presentations are a widely used and highly important medium for academic communication, and they are an essential part of lectures, seminars, tutorials, and conference talks (Hu & Wan, 2013). Creating a good slide deck from a scientific paper is time-consuming, demanding both content condensation with coherent narrative and layout design that keeps text–figure alignment across pages(Fu et al., 2022). Recent advances in multimodal models and LLM-based agents have motivated increasing efforts toward automating this process(Sun et al., 2021; Fu et al., 2022; Bandyopadhyay et al., 2024; Xi et al., 2025; Xu et al., 2025; Mondal et al., 2024; Shi et al., 2025; Zheng et al., 2025). Despite recent progress, two main gaps remain in slide generation: (i) prior work treat slide making as a compression task (Sun et al., 2021; Fu et al., 2022), but overlooks layout design and text–figure alignment; (ii) reference-clustering approaches to layout (Zheng et al., 2025) lack explicit control and visual feedback, yielding unstable, often low-quality results.

To move beyond summarization frameworks with little visual planning, we introduce **SlideGen**, a template-interpretable and modular framework that converts a scientific paper into a well-designed slide presentation. Figure 1 gives an overview of the proposed **SlideGen** framework. The pipeline begins with globally content parsing and asset extraction using docling (Livathinos et al., 2025). 1) **Outliner** prepares the slide plan by listing the sections and subsections, deciding how many slides each subsection contains, assigning textual content to each slide. 2) **Mapper** follows this plan to match the right figures and tables to the corresponding bullet points, and **Formulizer** locates equations and links the right formulas to the right bullets. 3) **Speaker** then turns the bullets into brief presenter notes and adds simple cues or transitions to move smoothly from one slide to the next. After that, **Arranger** selects a suitable template for each slide based on its planned content and mapped assets, then places and aligns all elements accordingly. 4) **Refiner** polishes the whole deck

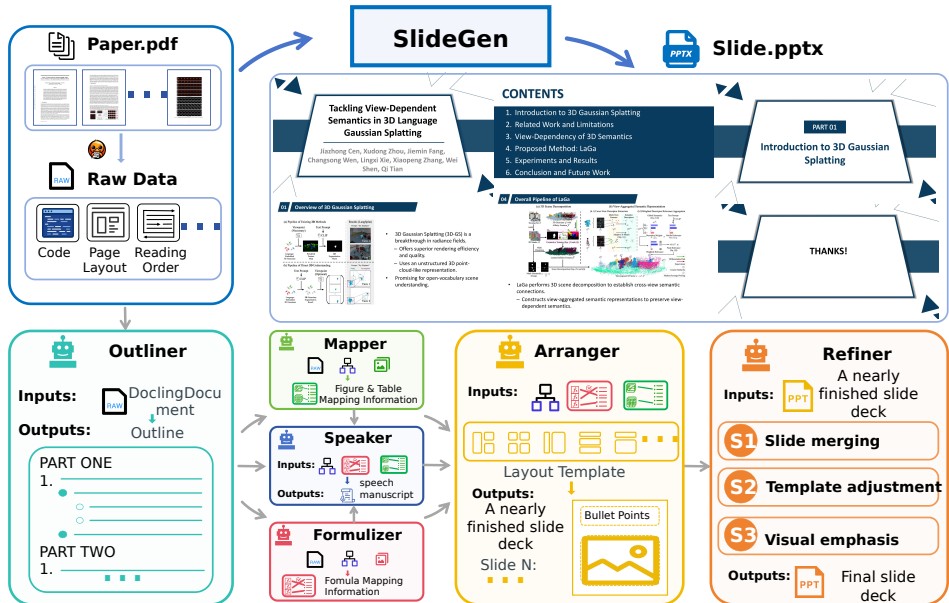

Figure 1: **Overview of the SlideGen pipeline.** The multi-agent frameworks consists of six agents that process a scientific paper in stages, including content planning, figure selection, layout design, formula explanation, visual refinement, and narration generation.

by merging slides with too little content, adjusting templates when needed, and adding moderate visual emphasis, such as bold, to make key points clearer.

To provide a clear basis for evaluating paper-to-slide generation, we introduce the **Paper2Slide Benchmark**. It consists of recent papers matched to high-quality slide decks, together with a standardized evaluation protocol that measures: (i) Visual Aesthetics – we use a geometry-aware density that rewards "just-right" layouts that are neither sparse nor cluttered, trading off target area against a moderate number of content regions; (ii) Communication Effectiveness – SlideQA measures how well the deck alone supports answering questions, using six VLM readers of varying capability, following (Pang et al., 2025); (iii) Holistic Assessment – VLM-as-Judge, a evaluation criteria over Content, Design and Coherence to provide a holistic view of deck quality, following prior work (Zheng et al., 2025); and (iv) Textual Coherence – the quality of expression in the slide presentation.

Using our **Paper2Slide benchmark**, we comprehensively evaluate state-of-the-art generative baselines (GPT-4o and GPT-5), and multi-agent methods, revealing several key findings: *(i)* the online GPT-4o/GPT-5 routes produce blurry, low-quality PPT images with few pages per answer, typically a single composite image covering only 4–9 slides, and it is impractical to output per-slide images or package them in a zip; GPT-4o is generally blurrier, while GPT-5 is clearer;*(ii)* GPT-5 is more prompt-sensitive and often fails to produce the requested outputs, while GPT-4o follows prompts more consistently; *(iii)* empirically, **SlideQA** correlates with human evaluation, and its scores increase with VLM capability on well-designed slides; and *(iv)* our **geometry-aware density** score separates sparse/balanced/cluttered layouts effectively as expected and aligns closely with human visual judgments. Our framework outlines practical paths toward next-generation end-to-end slide generation.

## 2 RELATED WORK

### 2.1 VISION–LANGUAGE AGENTS FOR SLIDES

Early document-to-slide systems framed the task as summarization (Sun et al., 2021; Fu et al., 2022; Kothawade et al., 2020) or sequence-to-sequence conversion from papers to slide decks. *D2S* casts slide generation as query-based single-document summarization(Sun et al., 2021). *DOC2PPT* introduces a sequence-to-sequence architecture with a learnable policy for section-to-slide progression

(Fu et al., 2022). Driven by VLM agents and multimodal learning (OpenAI et al., 2024; Naveed et al., 2025), recent work move from single-shot prompting to agentic, multi-stage pipelines for document-to-slides (Shi et al., 2025; Pang et al., 2025; Zheng et al., 2025). *DocPres* separates bird's-eye summarization, outline drafting, and slide-to-section grounding (Bandyopadhyay et al., 2024). As a multi-agent system, *RCPS* assigns clear, specialized roles: global planning via R-CoT, layout planning via LPG, and iterative refinement (Xi et al., 2025). Recent work also improves layout fidelity using a textual-to-visual "Reviewer+Refiner" loop (Xu et al., 2025). A representative high-performing approach, *PPTAgent* generates slides via a two-stage, edit-based pipeline using HTML layouts and self-correction, but it depends on references and may exhibit layout overlap or overflow, and also provides limited cross-page coherence (Zheng et al., 2025); in contrast, **SlideGen** is a visual-in-the-loop, multi-agent pipeline that plans globally, maps content precisely, composes layouts with an extensible template library, and keeps pages balanced rather than sparse or cluttered.

Among recent high-performing systems, *PPTAgent* adopts a two-stage, edit-based pipeline over HTML layouts with self-correction, yet it remains reference-dependent, prone to layout artifacts, including overlap and overflow, and limited in cross-page coherence (Zheng et al., 2025). In contrast, **SlideGen** performs end-to-end slide generation with explicit outline–layout grounding, precise figure/equation mapping, an extensible template library, and consistently balanced pages.

## 2.2 EVALUATION PROTOCOLS AND METRICS

Evaluation has evolved from text-only overlap (Sun et al., 2021; Fu et al., 2022) to multimodal and narrative-aware protocols (Pang et al., 2025; Zheng et al., 2025; Shi et al., 2025). Traditional automated metrics, including ROUGE (Lin, 2004) and perplexity (Jelinek et al., 1977), were used to measure slide text quality in early systems like *D2S* (Sun et al., 2021). *PresentAgent* (Shi et al., 2025) push beyond text by combining objective quizzes and subjective preferences, including two complementary axes: factual quizzes grounded in the source documents and preference-based vision–language scoring of presentation quality. VLM-as-judge (Bandyopadhyay et al., 2024; Zheng et al., 2025; Pang et al., 2025; Xi et al., 2025; Shi et al., 2025) has been adopted to rate overall slide quality, spanning content fidelity, design, and narrative coherence in recent work. However, prior metrics paid limited attention to the visual aesthetics of slides, and LLM-based scoring lacks theoretical grounding and reproducibility. We therefore propose Geometry-Aware Density, which provides a principled assessment of overall layout organization and aesthetics.

## 3 METHOD: A MULTI-AGENT FRAMEWORK FOR SLIDE GENERATION

**Overview**. SlideGen is a modular LLM-Agentic framework that transforms complete scientific papers into structured, readable, and well designed editable slides. Our agents begins with high-level content and structure planning, and proceeds step by step to detailed slide organization. It consists of six specialized agents, each responsible for a specific stage of the generation process, as shown in Fig. 1, In line with prior work (Pang et al., 2025), we first preprocess the raw PDF with DOCLING (Team, 2024) and MARKER (Paruchuri, 2025), converting pages to Markdown and assembling a two-modal asset library: *(i)* text assets capture the hierarchy by mapping section headings to brief, paragraph-level summaries like key–value pairs, and *(ii)* visual assets where figure and table captions index the extracted images.

**Outliner.** Outliner Agent converts a full research paper, provided as Markdown or plain text, into a two-level presentation outline optimized for slide-first delivery. Instead of copying original order of the paper, it reorganizes content toward an academic narrative and uses a recommended template: motivation & background, related work/limitations, key contributions, method overview, technical details, experiments & datasets, results & analysis, optional ablations/insights, and conclusion & future work. The agent applies the template case by case. For each paper, it may split long topics into part (a) and part (b) to keep sections focused, or merge weaker topics with adjacent sections to improve coherence. As a result, the final sectioning varies across papers rather than being identical. Outliner reads the entire document, identifies key ideas and dependencies, and produces a strict JSON object with two top-level keys: "metadata" and "sections" The "metadata" includes a title of the paper, the author name(s), the publication date, and the organization. The "sections" list follows the recommended flow above, with each section containing concise subsections that are ready for slide drafting, while page count and layout are left to downstream components. For example, when

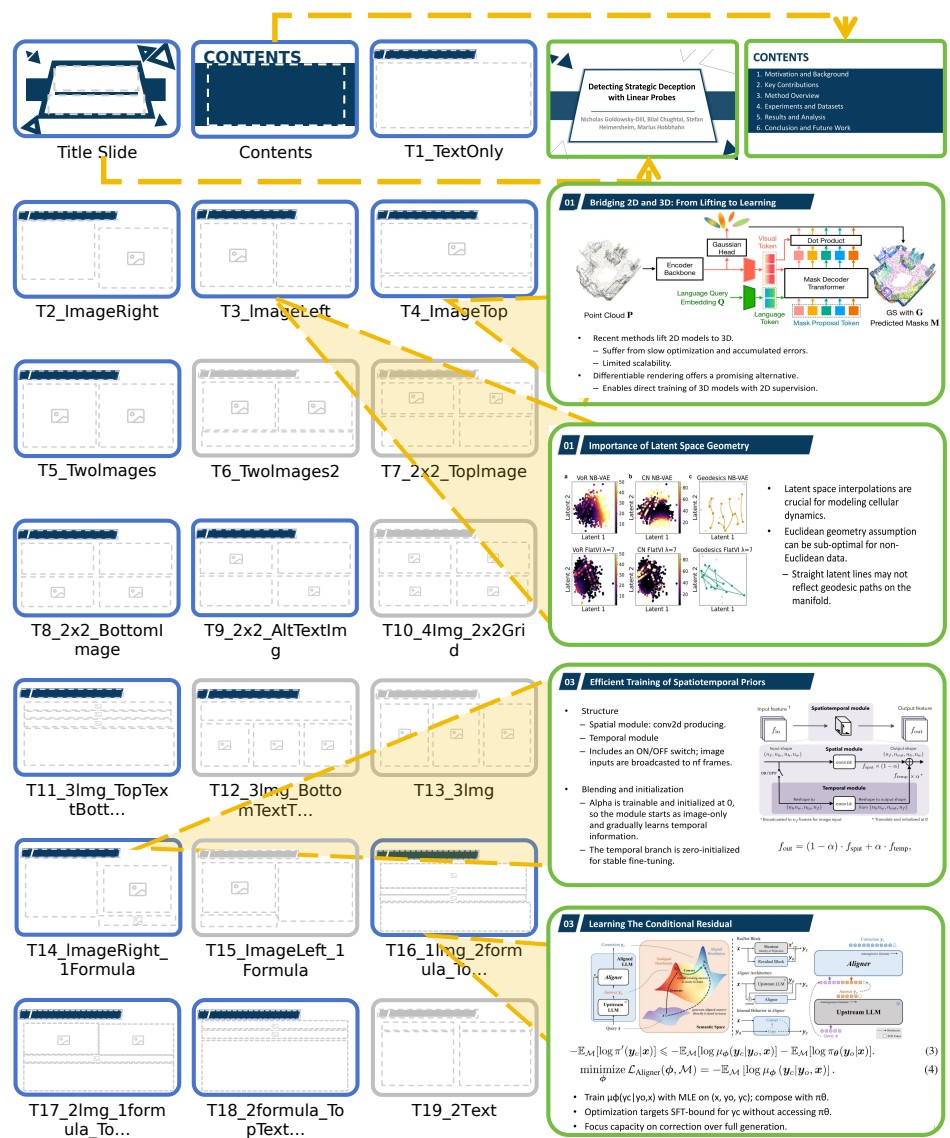

Figure 2: The slide template library used by the Arranger. Each template addresses a typical presentation structure, such as text-only, image-left, and two-column layouts.

Outliner processes a paper titled *ActionPiece*, it may propose the first two sections as *Introduction to Generative Recommendation* and *Proposed Method: ActionPiece*. The first section could include two subsections: *1.What Is Generative Recommendation?* and *2.Challenges in Current GR Models*. The second section would then continue with the method, potentially covering *Overview of ActionPiece*, *Vocabulary Construction*, and *Segmentation with Set Permutation Regularization*, and so on.

**Arranger.** While the Outliner determines what content goes into each slide, the Arranger decides how that content is presented. The Arranger is responsible for assigning an appropriate visual structure to each slide. This includes selecting a suitable layout template based on the amount and type of elements, the size and aspect ratio of a visual elements, and the overall balance between content and whitespace.

As shown in Figure 2, we design a small library of reusable slide templates that cover nearly all common presentation patterns. These include text-only layouts for background and conclusion slides, image-left or image-right layouts for highlighting key visuals, and layouts containing narrow strip-

like images of formulas, among others. For example, slides containing a prominent, wide-aspect image alongside a few sentences of text are very likely to be assigned to the T4 image-top template by Arranger. If the image is relatively tall or nearly square, the slide is more likely to be assigned to a half-and-half image–text template such as T2 or T3. By separating layout selection from content generation, the Arranger ensures slides are informative, visually balanced, and consistent with good presentation practice. It produces an almost complete deck, which is then handed to the Refiner for final adjustments.

**Refiner.** The Refiner improves the overall clarity and organization of the presentation. It performs several tasks: **(i) Slide merging.** Consecutive slides with very limited textual content without any visuals, are merged to reduce redundancy and maintain slide conciseness. **(ii) Template adjustment.** For the two consecutive text-only slides mentioned above, we switch their templates to T19_2Text. **(iii) Visual emphasis.** Important terms within bullet points are highlighted to guide attention. These improvements make the final presentation more engaging and easier to follow.

**Mapper.** The Mapper links figure and table assets to the pages it best supports. The Mapper produces a JSON file that, for each figure or table, lists the slide page it best supports and a brief reason. A single visual assets may be placed on multiple slides when appropriate, and not all assets must be used.

**Formulizer.** The Formulizer processes formula screenshots extracted from the paper. For each formula, it finds the most relevant section, writes a short explanation, and includes either the image or its LaTeX version. This helps preserve key mathematical content while making it easier to understand. We provide three methods for adding formulas: *(i)* Detect the bounding box coordinates of formulas and crop them directly. *(ii)* Extract the LaTeX code of formulas and render them. However, the rendered output may not always perfectly match the original formula, especially in terms of spacing, font, or stylistic nuances, leading to potential rendering crashes and errors. *(iii)* The user manually draws bounding boxes around the desired formulas in the prepared paper file. The framework then detects only the formulas within these user-defined regions. For each detected formula, the agent provides an interpretation and places it on the corresponding slide. This method ensures that only the user-selected formulas appear in the final presentation, making it the most content-precise approach. By default, we adopt method *(i)* as our primary approach throughout the pipeline.

**Speaker.** The Speaker creates a short narration script for each slide and directly reuses the placement rationales produced by the Mapper (for figures/tables) and the Formulizer (for equations), inserting them into the speaker notes. These scripts explain the key points in a clear and natural tone, and are stored in the note field of the slide. The presenter can use them directly or edit them as needed.

## 4 PAPER2SLIDE BENCHMARK

### 4.1 DATA CURATION

**Data Source.** We curated a domain-specific dataset focused on recent advances in machine learning and natural language processing, with a particular emphasis on research diversity and quality. Our dataset consists of 200 peer-reviewed papers collected from leading AI venues between 2022 and 2025, including only Oral presentations as designated by each conference. See in 1.

| Conference | 2022 | 2023 | 2024 | 2025 |
|------------|------|------|------|------|
| ICLR | 17 | 31 | 29 | 23 |
| ICML | – | 16 | 24 | 30 |
| NeurIPS | – | 10 | 20 | – |

Table 1: Number of papers by Conference and Year

These conferences were chosen for their rigorous review process, topical breadth, including multimodal learning, generative modeling, interpretability, and frequent inclusion of rich visual and mathematical content, making them ideal for downstream tasks such as slide generation, summarization, and modality-aware learning.

### 4.2 EVALUATION METRICS

We evaluate Paper2Slide Benchmark with four complementary metrics that jointly assess narrative quality, factual coverage, visual readability, and a quiz-style comprehension test.

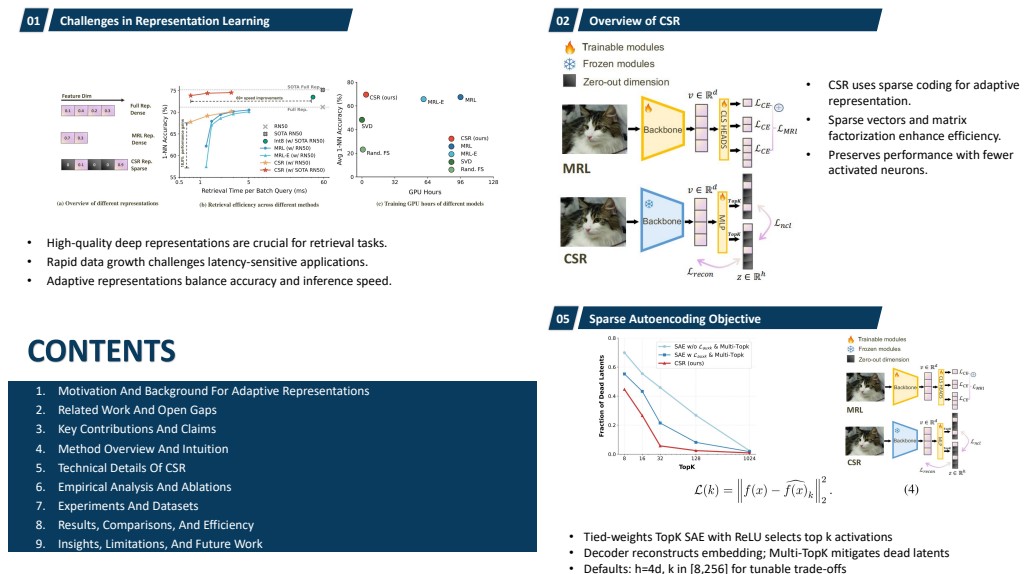

(a) SlideGen example outputs: the top row of slides is generated by GPT-4o, and the bottom row by GPT-5.

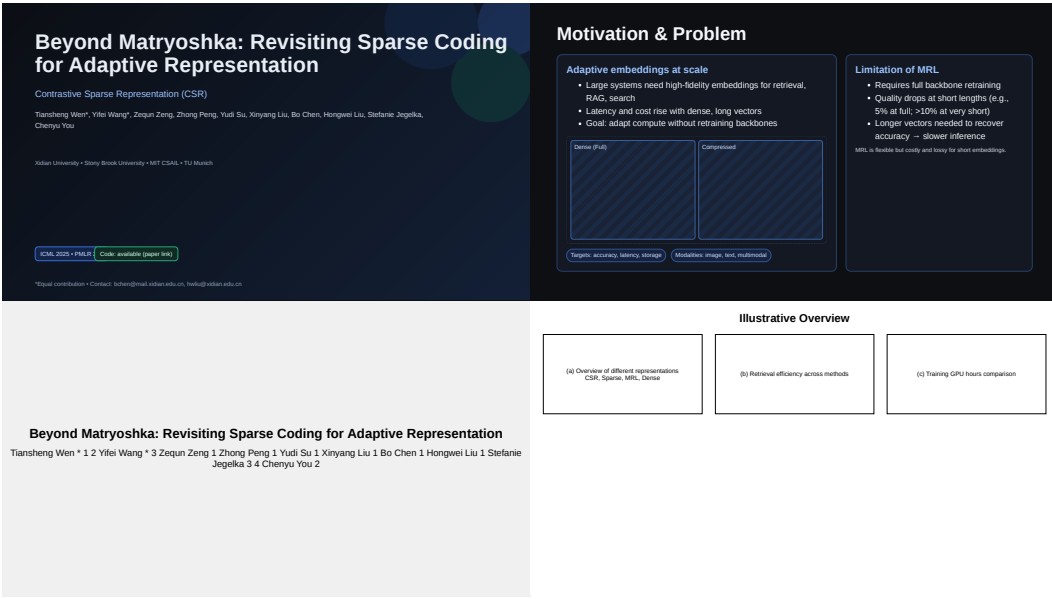

(b) GPT HTML example outputs: the top row of slides is generated by GPT-5 HTML, and the bottom row by GPT-4o HTML.

Figure 3: Comparative Analysis of Presentation Generation Methods

**Geometry-Aware Density.** We aim to quantify layout aesthetics and readability using mathematical function-based metrics. This metric evaluates layout density while also considering visually pleasing and comfortable design for human in terms of two components: *(i)* Area Occupancy: This measures how much of the slide's space is used, comparing it to a target occupancy value $\tau$. If the slide is too empty or too full, it negatively impacts the score. *(ii)* Effective Region Count: This measures the number of content regions on the slide. The ideal number of regions is denoted by $M^\star$, and the metric penalizes slides with too few or too many regions. The penalty is represented by a downward-opening quadratic function that rewards layouts with a number of regions close to $M^\star$. Overly blocky slides look rigid and lack hierarchy, while excessive partitioning introduces noise and jumpy reading. Details are provided in Appendix B.

**VLM as Judge.** Following PPTEVAL (Zheng et al., 2025), we evaluate decks along three dimensions – Content, Design and Coherence, using GPT-4o to judge. Scores are on a 1–5 scale, accompanied by brief rationales. The detailed criteria are listed in Table 5

**SlideQA.** Since slide decks are the primary vehicle by which speakers convey knowledge and audiences learn it, we need to evaluate whether our generated presentations communicate the material, and how much they succeed in doing so. Following PaperQuiz (Pang et al., 2025), for each paper, we first generate a quiz of 100 questions from the paper PDF: 50 verbatim questions answerable directly from the text, covering diverse factual aspects, and 50 interpretive questions targeting higher-level comprehension. Then the questions are answered by six different VLM readers, including three closed-source models: GPT-4o-mini, GPT-4o, and GPT-o3, and three open-source models: LLaVA-OV-7B, Qwen2-VL-7B-Instruct, and Phi-4-multimodal-instruct. The abilities of closed-source vision-language models typically surpass those of open-source models, akin to how more capable students demonstrate better overall learning abilities. To discourage verbosity, we apply a smooth length penalty to SlideQA with a calibrated coefficient so that average-length decks incur a target factor, details are provided in Appendix B.

**Textual Coherence.** Following the approach in (Pang et al., 2025), we quantify textual coherence using the standard "Perplexity" (PPL) metric, calculated for the entire poster text under Llama-2-7b-hf. A lower PPL score indicates more predictable and coherent language, see details in Appendix B.

## 5 EXPERIMENT

### 5.1 BASELINES AND SETTINGS

We evaluate our framework on multi slide PowerPoint generation with a fixed 16:9 canvas, the number of slides is unconstrained. The compared baselines span three categories: (i) *end to end generators*: GPT-5 HTML and GPT-4o HTML, which generate HTML+CSS code for slides, and GPT-5 Image and GPT-4o Image, which directly synthesize slide images page by page; (ii) *multi agent workflows*: PPTAgent-4o and PosterAgent-4o used in *slide mode*, which decompose planning, drafting, and layout into iterative editing steps; and (iii) *our method* instantiated with two backbones, GPT-4o and GPT-5, enabling a controlled comparison across backbones while keeping the rest of the pipeline unchanged.

All methods take the same source PDF per paper. We report *Length Penalized Accuracy* on SlideQA, distinguishing between Verbatim and Interpretive questions, and we categorize the models into open-source and closed-source groups, and provide separate evaluations for each group; overall *PPL* over concatenated slide text; and *Geometry Aware Density* with its two components, *Occupancy Match* and *Fragmentation Reward*; together with *VLM as Judge* scores along Content, Design and Coherence. Exact metric definitions and default thresholds are given in Sec. 4.2.

### 5.2 RESULTS

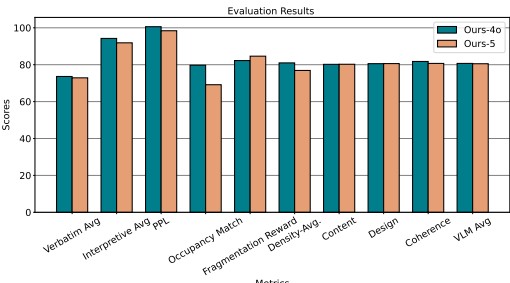

Figure 4: Performance of GPT-5 VS. GPT-4o in our benchmark

**Our method vs. baselines.** As shown in Table 2, Ours-4o delivers the strongest overall score in the table, improving over the best GPT-4o baseline, while maintaining very competitive interpretive performance without sacrificing verbatim coverage. This suggests our pipeline lifts detail retention without sacrificing global readability.

**Backbone Comparison and Stability.** Comparing our two backbones, Ours-4o outperforms Ours-5. GPT-5 demonstrates stronger coding ability but higher execution-failure and retry rates, and greater sensitivity to prompt phrasing—prompts that succeed

| Model | Verbatim ↑ | | | Interpretive ↑ | | | Overall Avg. |
|---|---|---|---|---|---|---|---|
| | open-src | closed-src | **V-Avg** | open-src | closed-src | **I-Avg** | |
| *GPT-5* | | | | | | | |
| HTML-5 | 74.89 | **70.27** | 72.58 | 91.22 | **90.92** | 91.07 | 81.83 |
| Image-5 | 66.93 | 53.94 | 60.44 | 73.13 | 89.49 | 81.31 | 70.87 |
| **Ours-5** | **75.73** | 70.03 | **72.88** | **94.30** | 89.41 | **91.86** | **82.37** |
| *GPT-4o* | | | | | | | |
| HTML-4o | 60.50 | 75.53 | 68.02 | 97.33 | 91.40 | 94.37 | 81.19 |
| Image-4o | 48.97 | 30.89 | 39.93 | 50.19 | 70.67 | 60.43 | 50.18 |
| PPTAgent-4o | 57.92 | 52.51 | 55.22 | 57.57 | 56.25 | 56.91 | 51.06 |
| PosterAgent-4o | 67.79 | 67.95 | 67.87 | 73.05 | 79.91 | 76.48 | 72.18 |
| **Ours-4o** | **75.93** | **71.32** | **73.63** | **94.67** | **93.82** | **94.25** | **83.94** |

Table 2: SlideQuiz Evaluation on SlideGen based on 6 different Readers.

with GPT-4o are more likely to be misinterpreted by GPT-5. We hypothesize this reflects a higher propensity for hallucination or overconfident, self-directed reasoning. We therefore tighten the system prompt and schema constraints. After iterative refinement, we identify a prompt that reliably yields valid GPT-5 outputs while preserving controllability.

**A persistent gap separates interpretive and verbatim accuracy in SlideQA.** Across all methods, *interpretive* accuracy is consistently and substantially higher than *verbatim* accuracy, as reflected in the SlideQA results reported in Table 2. This gap is large for most methods. The pattern indicates that fine-grained, quote-level details are harder to preserve and retrieve in multi-slide PPT generation than high-level understanding and reasoning. In practice this is expected: slides compress text, distribute content over multiple pages, and often replace long sentences with bullets or figures, thereby preserving the gist while reducing exact quote-level matches.

**HTML routes outperform image-only routes.** Using GPT to produce HTML/CSS significantly outperforms using it to produce pixel-based images. Image-only generation renders text as pixels, so it cannot be directly extracted and must rely on OCR. Because many "characters" are merely drawn, stroke-like approximations rather than standard glyphs, they often exhibit missing strokes, unintended joins, and distortions, which raise OCR error rates and further hinder content recognition; by contrast, HTML-based generation preserves actual text and layout structure, and the gap in readability and parseability between the two is substantial.

**Prompting Considerations for GPT-5.** In our pipeline, instruction fidelity differs noticeably between backbones. Empirically, GPT-4o follows schema-bound instructions with high Adherence: when asked to produce a plan as strict JSON, it reliably returns a well-formed object with the requested keys and structure. By contrast, a direct reuse of the same prompts on GPT-5 can yield schema violations. The most common failure mode is *mode collapse into a prose summary* instead of emitting the required JSON file, despite identical task intent. *Long system prompts tend to trigger a summarization mode.* This behavior suggests that prompt packaging, rather than task difficulty, is the dominant factor for GPT-5 under our setting.

A practical remedy is to design *backbone-specific* prompt packaging. When organizing prompts as YAML file with fields such as System Prompt, template, and jinja_args , we observe the following consistent pattern: keeping the System Prompt minimal and goal-focused, state only the task objective and moving the concrete requirements, including output schema, key lists, and formatting constraints, into `template` improves GPT-5's Adherence substantially. Conversely, a wordy System Prompt that mixes goals, checklists, and formatting often leads GPT-5 to *summarize* rather than to *conform to the requested output contract*. In practice, we therefore *(i)* keep the System Prompt to a single, unambiguous goal statement, and *(ii)* place the exact JSON schema, field-by-field constraints, and example scaffolds in the `template` block. Under this packaging, GPT-5's tendency to drift into summaries largely disappears, while GPT-4o continues to perform as before.

**Geometry-Aware Density** We decompose the Geometry-Aware Density into two components: Occupancy Match (OM) and Fragmentation Reward (FR). Across the benchmark, our approach achieves higher scores than all baselines on both OM and FR, indicating closer alignment to the

| Model | Perplexity | Density | | | VLM-as-Judge | | | |
|---|---|---|---|---|---|---|---|---|
| | | OM | FR | D-Avg. | Content | Design | Coherence | Avg. |
| *GPT-5* | | | | | | | | |
| HTML-5 | 189.38 | 54.29 | 60.75 | 57.52 | 3.54 | 4.02 | 4.09 | 3.88 |
| Image-5 | 605.02 | 67.98 | 79.39 | 73.69 | 2.84 | 3.16 | 3.21 | 3.07 |
| **Ours-5** | **98.40** | **69.15** | **84.62** | **76.89** | **4.12** | **4.30** | **4.35** | **4.26** |
| *GPT-4o* | | | | | | | | |
| HTML-4o | 200.79 | 41.19 | 46.46 | 43.83 | 3.02 | 2.76 | 3.97 | 3.25 |
| Image-4o | 793.71 | 70.29 | 76.20 | 73.25 | 2.39 | 3.09 | 3.50 | 2.99 |
| PPTAgent-4o | 721.54 | 53.22 | 56.26 | 54.74 | 3.25 | 3.24 | 3.29 | 3.26 |
| PosterAgent-4o | 139.67 | 68.73 | 76.20 | 72.47 | 3.19 | 3.48 | 4.53 | 3.73 |
| **Ours-4o** | **100.59** | **79.71** | **82.24** | **80.98** | **4.01** | **4.28** | **4.66** | **4.32** |

Table 3: Evaluation across Textual Coherence, Density (OM: Occupancy Match, FR: Fragmentation Reward, and weighted average D-Avg where $D\text{-Avg} = \lambda_1 \cdot \text{OM} + \lambda_2 \cdot \text{FR}$; default $\lambda_1 = 0.5, \lambda_2 = 0.5$), and VLM-as-Judge.

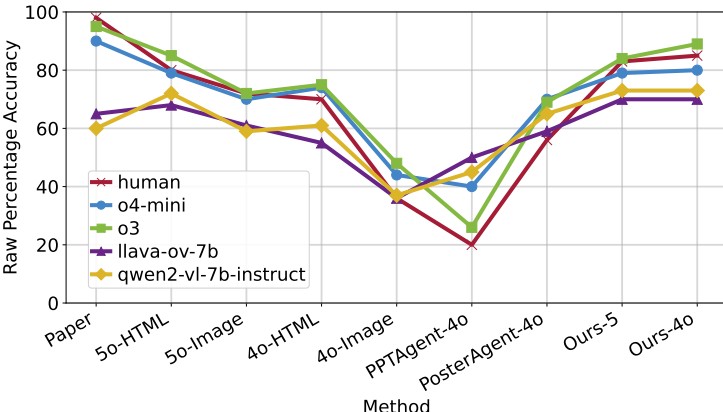

Figure 5: SlideQA's Avg scores across different types of slides (x-axis) for readers (colored lines) on human evaluation subset

target occupancy and a more effective region count near the preferred range M star. This indicates that, compared with those produced by the baselines, our generated PPT decks are neither sparse nor cluttered. Moreover, we observe that GPT-Image performs much better than GPT-HTML. This suggests that, even if the images are a bit blurry, GPT still aims for comfortable overall layout. In contrast, HTML is clear and precise, but the layouts often feel less comfortable and less appealing.

**Human evaluation.** To assess our method with human judgment, we recruited a PhD student to complete the SlideQA on 5 randomly selected papers from our **Paper2Slide** dataset, as shown in Table 5 For each paper, we evaluated 8 poster variants, including 6 baselines and 2 versions of our method, following the setup in Section 5.1. Details of the human evaluation protocol are provided in Appendix (). Figure 6 reports the average SlideQA scores per poster type (x-axis) for each reader (colored lines). Scores across poster types show good consistency between the human and the VLM readers. This alignment supports the use of reader models as effective proxies for human judgment.

## 6 CONCLUSIONS

We propose SlideGen, a step-by-step framework that covers outline planning, asset grounding, template selection, speaker-note drafting, and global refinement. We also introduce the Paper2Slide Benchmark with evaluation protocols for Geometry-Aware Density, VLM-as-Judge, SlideQA, and Textual Coherence. SlideGen advances automated slide generation toward human quality and improves efficiency, enabling practical, scalable scientific communication

**Ethics statement.** This work follows the ICLR Code of Ethics. We rely exclusively on publicly available datasets and pretrained models under their respective licenses. We do not anticipate direct negative societal impacts or ethical risks from the proposed method.

**Reproducibility statement.** We aim for complete reproducibility. All code, configuration files, and scripts required to replicate our experiments will be released publicly upon publication.

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

# Appendix

## A ABBREVIATIONS

We provide a reference for the abbreviations of models used in this paper.

| Abbreviation | Full Name |
|---|---|
| **4o-mini** | GPT-4o-mini |
| **4o** | GPT-4o |
| **o3** | GPT-o3 |
| **llava-ov-7b** | LLaVA-OneVision-Qwen2-7b-ov-hf (Li et al., 2024) |
| **Qwen2-VL-7B** | Qwen2-VL-7B-Instruct (Wang et al., 2024; Bai et al., 2023) |
| **Phi-4-MM** | Phi-4-multimodal-instruct (Abouelenin et al., 2025) |

Table 4: Reference for model abbreviations used in this paper.

## B METRIC DEFINITIONS AND PROTOCOLS

**Notation.** A deck consists of $N$ slides $\{s_i\}_{i=1}^N$. Each slide has a role $r_i \in \{\texttt{title}, \texttt{agenda}, \texttt{content}, \texttt{thanks}\}$. For $\texttt{content}$ slides we record an optional *section* label $\sigma_i \in \Sigma$ and *subsection* label $\sigma_i' \in \Sigma'$. We denote the pattern identifier by $\pi_i \in \mathcal{P}$ (e.g., $\texttt{T1\_TextOnly}$, $\texttt{T4\_ImageTop}$).

We consider a fixed slide layout: slide $s_1$ is the $\texttt{title}$ page, slide $s_2$ is the $\texttt{agenda}$ page, slides $s_3, \ldots, s_{N-1}$ are $\texttt{content}$ pages, and the last slide $s_N$ is $\texttt{thanks}$ page. Formally, a deck has $N$ slides $\{s_i\}_{i=1}^N$ with roles $r_1 = \texttt{title}$, $r_2 = \texttt{agenda}$, $r_i = \texttt{content}$ for $3 \le i \le N-1$, and $r_N = \texttt{thanks}$. The content page lists *section dividers* ("PART 1, PART 2, …"); these are the *agenda items*. Let $\mathcal{A} = [a_1, \ldots, a_m]$ be the ordered list of top-level bullets on $s_2$.

Each slide carries a hierarchical string bullet list $B_i = \{(u_{i,k}, \mathcal{S}_{i,k})\}_{k=1}^{K_i}$, where each content box $b$ is defined as a pair $(u_{i,k}, \mathcal{S}_{i,k})$, and $u_{i,k}$ is the $k$-th top-level bullet and $\mathcal{S}_{i,k} = [v_{i,k,1}, \ldots, v_{i,k,L_{i,k}}]$ is the list of sub-bullet strings.

We define the flattened textual content $\text{flat}(B_i) = [u_{i,1}, \mathcal{S}_{i,1}, \ldots, u_{i,K_i}, \mathcal{S}_{i,K_i}]$ and let $w_i = \text{words}(\text{flat}(B_i))$ be the word count. Image, table, and formula assets on slide $i$ are denoted by the finite sets $\mathcal{I}_i$ for image filenames, $\mathcal{T}_i$ for table filenames, and $\mathcal{F}_i$ for LaTeX strings, respectively. Optional speaker notes are written $n_i$.

Let slide area be 1. For each region $b \in \mathcal{B}_i$ with normalized width and height $w_b, h_b$, its area is $A(b) = w_b h_b$. The occupied area is the union area $\rho_i \in [0, 1]$ of all non-background regions.

### B.1 GEOMETRY-AWARE DENSITY

This metric evaluates layout density with two components: (i) area occupancy relative to a target $\tau$; (ii) a concave quadratic preference over the effective number of content boxes, peaking at $M^\star$.

**Why a downward-opening scoring function?** Overly monolithic slides look blocky and lack hierarchy, while excessive partitioning introduces noise and jumpy reading. A downward-opening scoring function over the effective region count captures the optimal range: it peaks near the preferred count $M^\star$, then smoothly decreases as the count drifts left, where pages become too plain, or right, where they become too busy, avoiding brittle thresholds. The width $\kappa$ controls tolerance around $M^\star$, and the area gate $a_{\min}$ prevents gaming with tiny micro-regions. Combined with the occupancy term $1 - |\rho_i - \tau|$, this yields an interpretable and reproducible measure that rewards layouts which are neither sparse nor cluttered.

We count only non-trivial regions via a minimum area gate $a_{\min} > 0$:

$$M_i^{\text{eff}} = \sum_{b \in \mathcal{B}_i} \mathbf{1}[A(b) \ge a_{\min}]. \tag{1}$$

Define a downward-opening quadratic fragmentation reward with maximum at $M^\star$:

$$R_i^{\text{frag}} = \max\left\{ 0,\ 1 - \frac{(M_i^{\text{eff}} - M^\star)^2}{\kappa^2} \right\} \in [0, 1]. \tag{2}$$

**OM and FR decomposition.**

$$\text{OM}_i \triangleq 1 - \left| \rho_i - \tau \right|, \qquad \text{FR}_i \triangleq R_i^{\text{frag}}. \tag{3}$$

$$s_i^{\text{geom}} = \lambda_1 \, \text{OM}_i + \lambda_2 \, \text{FR}_i, \qquad \lambda_1 + \lambda_2 = 1, \tag{4}$$

$$\text{DENSITY}^{\text{geom}} = \frac{1}{N} \sum_{i=1}^{N} \left( \lambda_1 \, \text{OM}_i + \lambda_2 \, \text{FR}_i \right). \tag{5}$$

We set $a_{\min} = 0.04,\ M^\star = 3,\ \kappa = 2.1,\ \tau = 0.55,\ \lambda_1 = 0.6,\ \lambda_2 = 0.4$.

## B.2 PPTEVAL CRITERIA

| Dimension | Criteria |
|---|---|
| Content | Text is concise and grammatically sound; key points are supported by relevant images. |
| Design | Harmonious colors and proper layout ensure readability; visual elements enhance appeal without clutter. |
| Coherence | Structure progresses logically and includes essential background information across the deck. |

Table 5: PPTEVAL dimensions and criteria (1–5 scale), adapted from (Zheng et al., 2025).

### B.2.1 SLIDEQA PROTOCOL

The protocol of SlideQA is as follows: **(i) Question curation:** For each source paper, we follow a poster–reader communication setup (Pang et al., 2025) and employ ChatGPT-5o as a question-generation model to produce $|\mathcal{Q}_{\text{eval}}| = 100$ multiple-choice questions per paper. We construct two disjoint subsets: $\mathcal{Q}_{\text{verb}}$ with $|\mathcal{Q}_{\text{verb}}| = 50$ *verbatim* questions directly answerable from the paper text, spanning 13 content aspects; and $\mathcal{Q}_{\text{int}}$ with $|\mathcal{Q}_{\text{int}}| = 50$ *interpretive* questions targeting high-level comprehension across 10 conceptual dimensions. We set $\mathcal{Q}_{\text{eval}} = \mathcal{Q}_{\text{verb}} \cup \mathcal{Q}_{\text{int}}$ and $\mathcal{Q}_{\text{verb}} \cap \mathcal{Q}_{\text{int}} = \varnothing$. **(ii) Respondents:** Each image is presented to $M = 3$ vision–language models, a mix of open- and closed-source systems, to simulate reader standards from casual to expert (Pang et al., 2025). Each model answers all $|\mathcal{Q}_{\text{eval}}|$ questions using only the poster content.

**Definition.** Let $r_{q,m} \in \{0, 1\}$ denote the correctness of model $m \in \{1, \ldots, M\}$ on question $q \in \mathcal{Q}_{\text{eval}}$. Define the per-question averaged correctness

$$\bar{r}_q = \frac{1}{M} \sum_{m=1}^{M} r_{q,m}. \tag{6}$$

The SlideQA accuracy is then

$$s_R = \frac{1}{|\mathcal{Q}_{\text{eval}}|} \sum_{q \in \mathcal{Q}_{\text{eval}}} \bar{r}_q, \tag{7}$$

which averages correctness across both questions and models. Subset scores restrict the sum in equation 7 to $\mathcal{Q}_{\text{verb}}$ and $\mathcal{Q}_{\text{int}}$:

$$s_R^{\text{verb}} = \frac{1}{|\mathcal{Q}_{\text{verb}}|} \sum_{q \in \mathcal{Q}_{\text{verb}}} \bar{r}_q, \qquad s_R^{\text{int}} = \frac{1}{|\mathcal{Q}_{\text{int}}|} \sum_{q \in \mathcal{Q}_{\text{int}}} \bar{r}_q. \tag{8}$$

**Rationale.** This protocol simulates how readers glean information from posters: questions come from the paper, but answers must be inferred solely from the poster. To avoid rewarding text-heavy decks, we additionally provide a length-penalized variant $s_R^{\text{LPA}}$ via the adjustment in Appendix B.3.

### B.3 Length-Penalized Accuracy (LPA)

**What it measures.**   It combines raw QA accuracy with a length penalty so that equally accurate yet shorter decks receive higher scores.

**Why LPA?**   LPA discourages decks that chase QA accuracy by copying long passages and instead rewards concise slides that communicate the core ideas clearly.

**Definition.**   Let the total deck length be

$$l = \sum_{i=1}^{N} \text{tok}\big(\text{flat}(B_i)\big), \tag{9}$$

where $\text{tok}(\cdot)$ returns the token count under a fixed tokenizer. We define the length-penalized score

$$\text{LPA}(\alpha) = \frac{s_R}{1 + \alpha \cdot \log(1 + l)}, \qquad \alpha > 0, \tag{10}$$

which is bounded by $\text{LPA}(\alpha) \leq s_R \leq 1$ and decreases smoothly as $l$ grows.

**Calibration.**   To set the penalty strength, we choose $\alpha$ so that the denominator in equation 10 at the average length $\bar{l}$ equals a target factor $1 + \gamma$ ( $\gamma \in [0.2, 0.5]$):

$$\alpha = \frac{\gamma}{\log(1 + \bar{l})}. \tag{11}$$

### B.4 Perplexity (PPL)

**What it measures.**  It quantifies the average next-token uncertainty of a language model over the deck text. Lower values indicate more fluent and predictable text. We compute this metric using Llama-2-7b-hf language model.

**Definition.**   Let $T(\cdot)$ be a fixed tokenizer and let

$$x_{1:L} = T\Big(\text{flat}(B_1) \parallel \cdots \parallel \text{flat}(B_N)\Big)$$

be the token sequence obtained by concatenating all slide texts. The full-sequence perplexity is

$$\text{PPL} = \exp\left(-\frac{1}{L}\sum_{t=1}^{L}\log p_\theta\big(x_t \,\big|\, x_{<t}\big)\right), \tag{12}$$

where $\log$ denotes the natural logarithm. Lower PPL means higher predicted likelihood per token; $\text{PPL} = 1$ corresponds to perfectly predictable text.

## C SlideGen Benchmark dataset

Here we present the **Paper2Slide** Benchmark, our generated dataset. Representative samples are shown in 6, 7, 8, 9, 10

## D Prompts

We provide the prompts used in our framework and benchmark for reference, see Figs. 11,12,13,14,15,16 and 17.

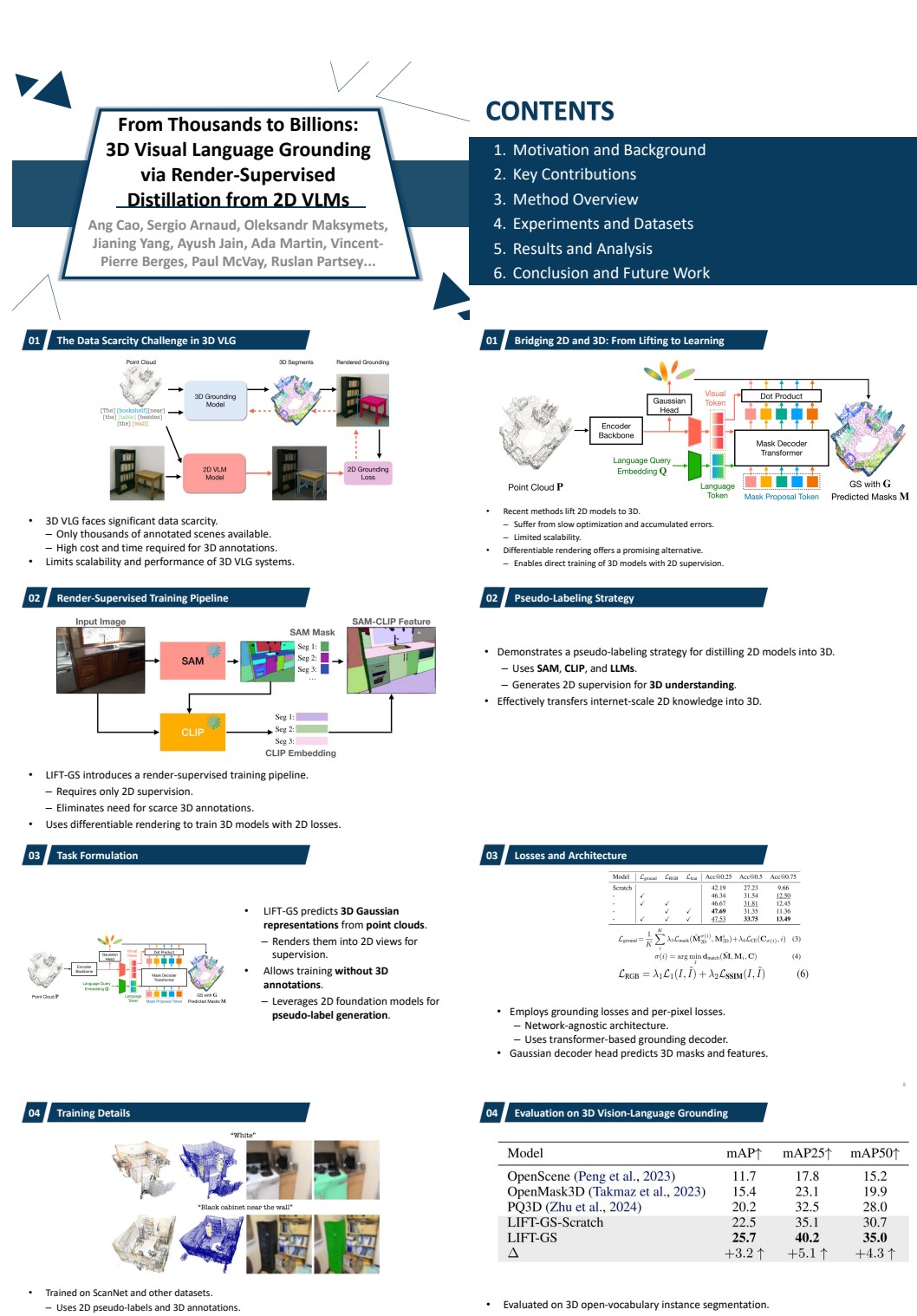

Figure 6: Representative sample 1 from the Paper2Slide Benchmark.

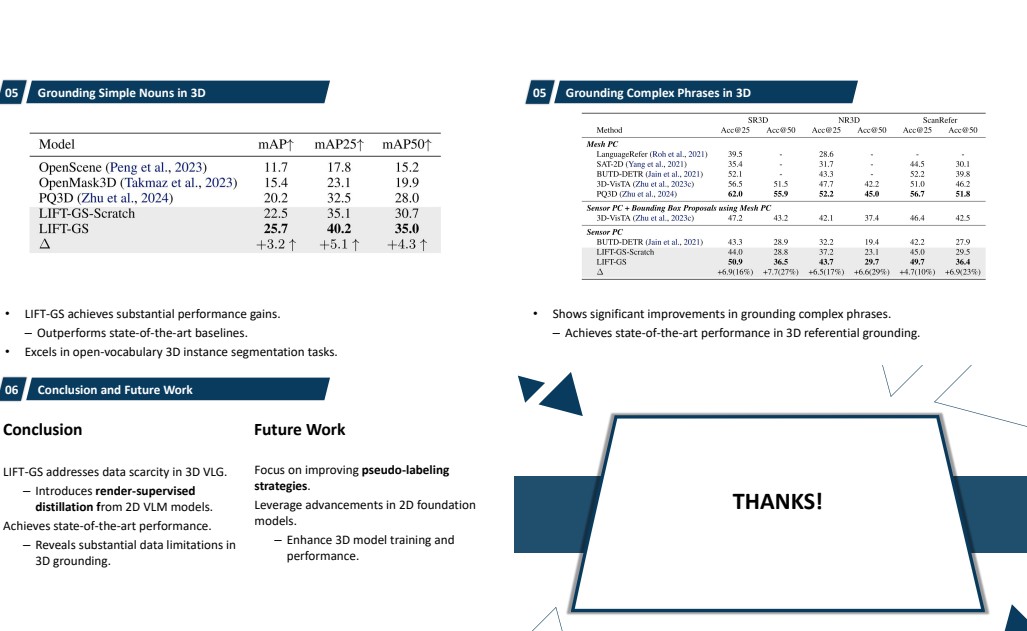

Figure 7: Representative sample 1 from the Paper2Slide Benchmark.

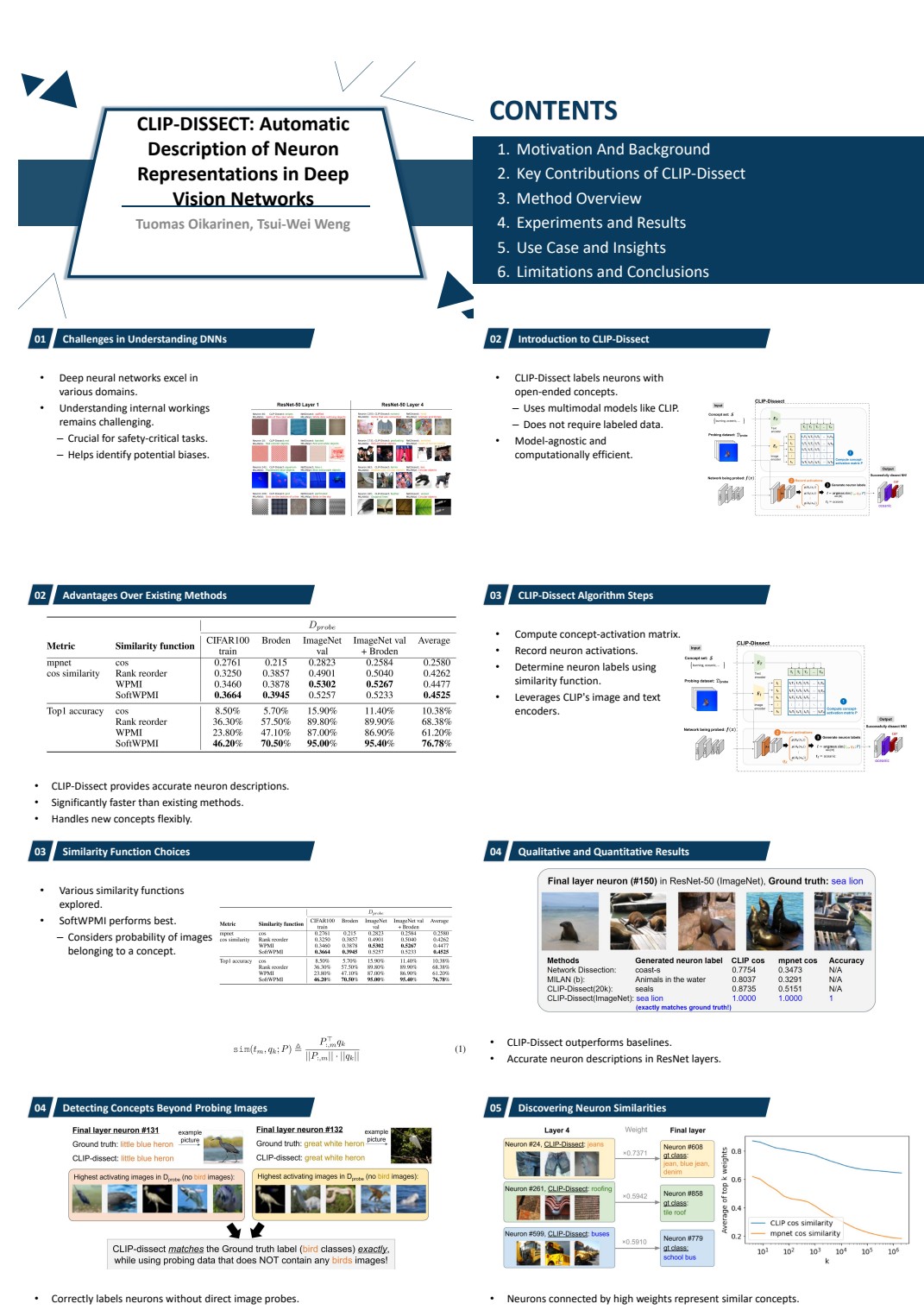

Figure 8: Representative sample 2 from the Paper2Slide Benchmark.

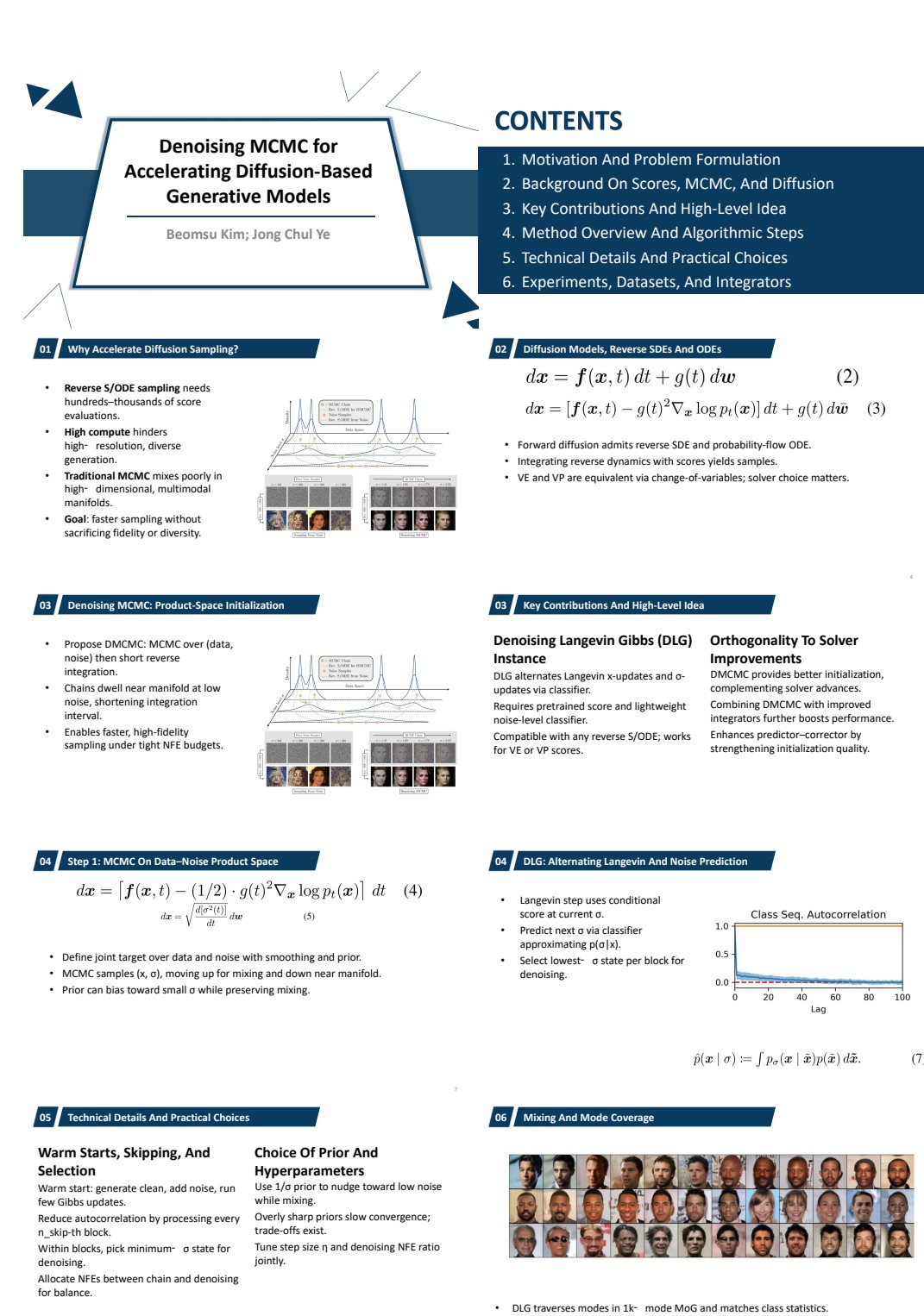

Figure 9: Representative sample 3 from the Paper2Slide Benchmark.

**06 Image Generation Benchmarks**

- DLG accelerates multiple samplers across CIFAR‑10, CelebA‑HQ, FFHQ.
- Reduces NFEs needed for competitive or better FID.
- Works for deterministic and stochastic integrators.

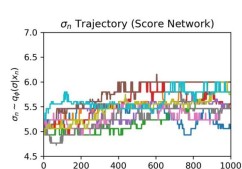

**06 Conditional Generation And Scores**

| Class | 0 | 1 | 2 | 3 | 4 | 5 | 6 | 7 | 8 | 9 |
|-------|------|------|------|------|------|------|------|------|------|------|
| No DLG | 14.3 | 11.6 | 15.8 | 17.7 | 14.7 | 16.9 | 16.0 | 13.4 | 11.1 | 11.3 |
| WIth DLG | 12.2 | 9.3 | 13.5 | 14.8 | 11.6 | 13.6 | 12.7 | 10.6 | 9.3 | 8.5 |

- DLG improves class‑conditional generation with VE and VP scores.
- Per-class FID improves when adding DLG to same integrator.

**07 State-Of-The-Art In Low-NFE Regime**

| Method | NFE 10 | NFE 20 | NFE 50 |
|--------|--------|--------|--------|
| DPM-Solver-2 (VP) | 5.28 (+2 NFE) | 3.02 (+4 NFE) | 2.69 (−2 NFE) |
| DPM-Solver-3 (VP) | 6.03 (+2 NFE) | 2.75 (+4 NFE) | 2.65 (−2 NFE) |
| DEIS (VP) | 4.17 (+0 NFE) | 2.86 (+0 NFE) | 2.57 (+0 NFE) |
| DEIS (VE) | 20.89 (+0 NFE) | 16.59 (+0 NFE) | 16.31 (+0 NFE) |
| KAR1 (VP) | 9.70 (+1 NFE) | 3.23 (+5 NFE) | 2.97 (+1 NFE) |
| KAR1 (VE) | 14.12 (+1 NFE) | 4.46 (+5 NFE) | 4.1 (+1 NFE) |
| DLG+KAR1 (VP) | **3.25** (+0.1 NFE) | **2.49** (−3.9 NFE) | 2.49 (−33.9 NFE) |
| DLG+KAR1 (VE) | 3.86 (+0.1 NFE) | 2.63 (+0.1 NFE) | **2.45** (−0.9 NFE) |

- DLG+KAR1 achieves SOTA FID at ~10–16 NFE on CIFAR‑10.
- CelebA‑HQ‑256: DLG+KAR2 outperforms prior 4000‑NFE results.
- FFHQ‑1024 shows large low‑NFE FID gains.

**07 Ablations: η, NFE Split, And Necessity Of Denoising**

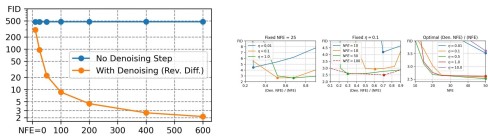

- Optimal η and denoising‑to‑total NFE ratio balance diversity and quality.
- As NFE grows, near‑optimal ratios widen.
- Removing denoising collapses quality—denoising is essential.

**07 σ-Trajectory And Manifold Proximity**

- σ trajectories move up/down, enabling mode transitions.
- Predicted σ correlates with distance‑to‑manifold scaling.
- Classifier keeps chains where score gradients are informative.

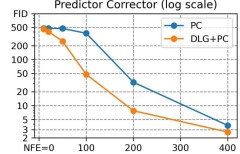

**08 Relation To Predictor-Corrector And Distillation**

- DMCMC complements PC by improving initialization; accelerates PC pipelines.
- Compared to distillation, requires far less extra training compute.
- Achieves competitive FID at similar NFE with minimal overhead.

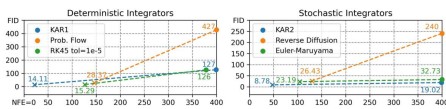

**08 Related Work, Limitations, And Impact**

**Limitations And Future Extensions**

Extensions to guided diffusion (classifier/CLIP) are natural next steps.

Further theory on Langevin Gibbs convergence and adaptive priors needed.

Trade-offs between stability and speed warrant deeper analysis.

**Societal Impacts And Reproducibility**

Acceleration reduces compute and energy for generative models.

Faster sampling can amplify misuse risks; responsible deployment is needed.

Code and checkpoints provided with clear hyperparameters and pseudocode.

**09 Main Takeaways**

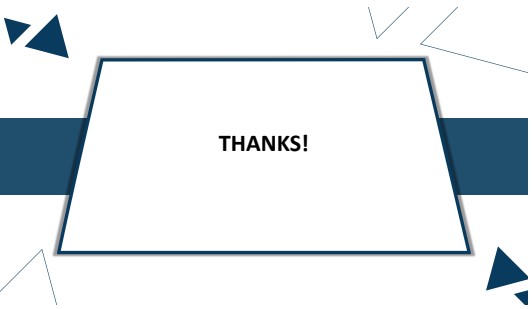

- DMCMC samples in **data–time space** first, then denoises, shortening integration.
- DLG is simple, plug‑and‑play, and scales to high resolution.
- Delivers state‑of‑the‑art results in low‑NFE regimes.

**THANKS!**

Figure 10: Representative sample 3 from the Paper2Slide Benchmark.

1080
1081
1082
1083
1084
1085
1086
1087
1088
1089
1090
1091
1092
1093
1094
1095
1096
1097
1098
1099
1100
1101
1102
1103
1104
1105
1106
1107
1108
1109
1110
1111
1112
1113
1114
1115
1116
1117
1118
1119
1120
1121
1122
1123
1124
1125
1126
1127
1128
1129
1130
1131
1132
1133

## layout_agent_xin

system_prompt:
  You are SlidePlanBuilder.
  Your ONLY task: return a single valid JSON object matching EXACTLY the schema below.
  Do NOT include explanations, summaries, markdown code fences, or natural language.

template:
  Instructions:
  The PowerPoint canvas is **fixed at 13.3 in* 7.5 in** (16:9).You receive five JSON blobs:
  1. **raw_result.json** - hierarchical summary of the paper.  Structure:
  2. **figures.json** - list of sections → subsections → visual assets.  Example (keys may vary by paper):
    *Each `imageN` or `tableN` value is an index that maps to an image/table file name (`image_2.png`, `table_1.png`, etc.).*
  3. **formula_index.json** - flat list of formula images:
  4. **image_dims.json**  - pixel dimensions for every `image_.png`
  5. **table_dims.json**  - pixel dimensions for every `table_.png`

  What you must do for **every subsection**

  1. **Pick the best slide template** from this library and output its `template_id`:

  | ID | When to use |
  |----|-------------|
  | T1_TextOnly   | No images/tables |
  | T2_ImageRight | 1 image + ≤4 bullets |
  | T3_ImageLeft  | Mirror of T2 (alternate left/right across consecutive slides) |
  | T4_ImageTop   | 1 wide image (aspect > 1.6) or table |
  | T5_TwoImages        | Exactly 2 side-by-side images, no text                     |
  | T5_TwoImages2        | Two side-by-side images on top, with a text block below            |
  | T7_2x2_TopImage       | 2*2 layout: top two blocks are images, bottom two are text               |
  | T8_2x2_BottomImage     | 2*2 layout: top two blocks are text, bottom two are images               |
  | T9_2x2_AltTextImg     | 2*2 layout: images on top-left & bottom-right, text on top-right & bottom-left |
  | T10_4Img_2x2Grid       | Four images arranged in a 2*2 grid, no text                      |
  | T11_3Img_TopTextBottom | Vertically divided: 3 images on top, text block below            |
  | T12_3Img_BottomTextTop  | Text block on top, 3 square images in one row below             |
  | T13_3Img           | Title on top, followed by 3 evenly spaced images                 |
  | T14_ImageRight_1Formula | Right column has two slots: top-right = one image or one table, bottom-right = one formula; left column = text bullets. Use when the slide has one key equation plus one main visual. |
  | T15_ImageLeft_1Formula | Left column has two slots: top-left = one image or one table, bottom-left = one formula; right column = text bullets. Use when the slide has one key equation plus one main visual.  |
  | T16_1Img_2formula_TopTextBottom | Bottom = text block; top are three rows: row1 = one

image or one table, row2 = one formula, row3 = one formula. Use for one main visual plus two formulas. |
| T17_2Img_1formula_TopTextBottom | Top row: two visuals side by side (each is one image or one table); middle row: one formula; bottom: text block. |
| T18_2formula_TopTextBottom | Top 2 rows: two formulas; bottom: text block. |

2. **Generate hierarchical bullets** summarising the subsection:
   • Up to **6 top-level bullets**.
   • Each top bullet may have **0-6 sub-bullets** (2-level outline).
   • Top bullets ≤ 20 words; sub-bullets ≤ 25 words.

3. **Select visuals** that best support the bullets:
   • **Formulas** belonging to the same subsection should stay **on the same slide whenever possible**; if more than 2, prefer `T11_3Img_TopTextBottom`.
   • **Do not crop or distort images** - preserve original aspect ratio (minor scaling to fit is fine).

4. **Return a single valid JSON object** with the exact schema below - do **NOT** wrap it in markdown.
   ```json

```
{
   "slides": [
     {
       "section": "<string>",
       "subsection": "<string>",
       "template_id": "T?_",
       "bullets": [
         {
           "text": "<string>",
           "sub": ["<string>", ...]
         }, ...
       ],
       "images": ["<filename>", ...],
       "tables": ["<filename>", ...],
       "formulas": ["<filename>", ...]
     }, ...
   ]
}
```

*Use the template-selection rules strictly so that downstream code can rely on them.*
Answer **only** with the JSON.

You **must** consider each visual's size and aspect ratio

*For every image / table, compute aspect = width ÷ height.*
*Choose the slide template and left/right/top placement based on aspect and absolute size:*
 *- **Wide** (aspect ≥ 1.6) → best placed across the top (template **T4_ImageTop**), including wide tables.*
 *- **Tall / square** (aspect ≤ 1.0) → best placed on the left or right (templates **T2_ImageRight** or **T3_ImageLeft**).*
 *- If a visual's width is nearly the full slide width, prefer **T4_ImageTop** to avoid excessive down-scaling.*
 ***Never** stretch or crop; only scale proportionally to fit placeholders.*

Figure 11: Prompt for Arranger.

When designing slide layouts, you must carefully consider visual density and legibility constraints—especially for images that are wide or contain fine-grained details.

Such images often become unreadable when downscaled to fit dual-visual layouts like T2_ImageRight, T3_ImageLeft, or T5_TwoImages2.

If multiple visuals(such as two images both with an aspect ratio greater than 1.6) are assigned to the same subsection but combining them would result in overcrowding or poor legibility, first check whether one of them fits better semantically in a neighboring subsection (e.g., covering a related topic or dataset). If so, move it to that subsection and assign a layout that presents it alone.

```
raw_result:
{{ raw_result_json }}
figures:
{{ figures_json }}
  formulas:
{{ formulas_json }}
image_informations:
{{ image_informations_json }}
table_informations:
{{ table_informations_json }}

jinja_args:
 - raw_result_json
 - figures_json
 - formulas_json
 - image_informations_json
 - table_informations_json
```

Figure 12: Prompt for Arranger.

```
formula_match

system_prompt: |
  You are an expert assistant tasked with assigning formulas to the most relevant paper sections.
  You will be given:
    1. JSON content of the paper structure, including sections and subsections (with title and
  description).
    2. A list of formulas with LaTeX, page_no, and the surrounding text context.
  GOAL:
    • Each formula should be assigned to its corresponding subsection, and a subsection may contain
  multiple formulas.
    • Produce a new JSON object that mirrors the structure of the provided paper_outline_json
  (sections → subsections).
    • For each subsection, assign zero, one, or multiple formulas.

    • For each assigned formula, include:
        - "formulaN": <formula_id>
        - "reasonN": <reason string> explaining why it's assigned
    • For each formula assigned to a subsection, generate a reason string ("reasonN") that not only
  explains why the formula is assigned to this specific subsection,
       but also briefly interprets the formula's mathematical meaning or role within the paper.
    • A formula may be assigned to multiple subsections (if conceptually appropriate), but not multiple
  times in the same subsection.
    • Keys must use correct suffixing: formula, formula1, formula2,... and reason, reason1, reason2,...
    • Keep section/subsection titles exactly as-is. Do not include their full content in the output.
    • The final result should be a single valid JSON structure.
  THINKING STRATEGY:
    • Use the surrounding context and page_no from the formula list to guide assignment.
    • Match concepts using keywords, notation, or nearby words (e.g., if the section talks about
  "posterior", and the formula mentions p(x|y), that's a match).
    • Try to ensure each early-indexed formula (e.g. formula 1-5) is assigned at least once.
    • Do not assign arbitrarily.

  OUTPUT FORMAT:
  {
    "sections": [
      {
        "title": "<Section Title>",
        "subsections": [
          {
            "title": "<Subsection Title>",
            "formula1": <id>,
            "reason1": "<explanation>",
            "formula2": <id>,
            "reason2": "<explanation>"
          },
          ...
        ]
      },
      ...
    ]
}
```

Figure 13: Prompt for Formulizer.

```
  }
CAUTION:
  - Output must be valid JSON only (no comments or explanations).
  - Only include sections/subsections where at least one formula is assigned.
  - Match titles exactly from the original input.
template: |
  Instructions:
  1. Analyze the paper outline: {{ json_content }}
  2. Analyze the list of formulas with their latex and context: {{ formula_information }}
  3. For each subsection, decide which formulas (if any) are conceptually relevant based on content
and wording.
  4. Match carefully using terms, equations, symbols, and latent meaning.
  5. Output a single JSON object following the system_prompt rules.
jinja_args:
  - json_content
  - formula_information
```

Figure 14: Prompt for Formulizer.

**figure_match**

system_prompt: |
  You are an expert assistant tasked with assigning images and tables to the most relevant paper sections.
  You will be given:
    1. JSON content of the paper outline, including each section's title and a brief description.
    2. A list of images (image_information) with captions and size constraints.
    3. A list of tables (table_information) with captions and size constraints.

  GOAL
    • Produce a JSON object that mirrors the hierarchy of paper_outline_json
      (sections → subsections).
    • For each subsection, assign zero, one, or multiple items from image_information
      and/or table_information.
    • Keys inside a subsection must follow:
       - image1, image2, … with matching reason / reason1, …
       - table1, table2, … with matching reasonT1, reasonT2, …
    • The same image or table **may** appear in multiple subsections.
    • Ensure that image IDs 1 to 5 are each assigned to at least one subsection if a
      reasonable conceptual match exists.
    • If multiple images or tables match a section well, include all of them. Assign each item only once
  per section, using different keys: e.g., "image", "image1", "table", "table1", etc.
    • If assigning an image, specify "image": <id>, where <id> is the identifier of the chosen image
  from "image_information".
    • If assigning a table, specify "table": <id>, where <id> is the identifier of the chosen table from
  "table_information".
    • Include an additional "reason", "reason1", etc. field briefly explaining why this assignment was
  made (e.g., how the image/table relates to the section content).
    • If no image or table is assigned to a given section, omit that section from the final JSON (i.e.,
  only list sections where you actually assign something).
    • Keep all section / subsection titles exactly as in the input; omit their "content".

  IMPORTANT:
    • The assignment should not be arbitrary. It must be logically consistent with the section's
  description and the provided caption for the image or table.
    • Do not produce any layout properties or subsections here.
    • The final output must be a single JSON object, mapping from section names to the chosen
  image/table ID plus the "reason" field.
    • Extra note: If multiple images or tables are suitable, select the single best one and assign only
  that.
    • If "image_information" or "table_information" is empty, you may end up assigning nothing to
  any section.

template: |
  Instructions:
    1. Read and analyze the paper's sections from {{ json_content }} .
    2. Look at {{ image_information }} and {{ table_information }}. Determine content-fit:
      - If a section's description or subject matter matches well with a given image/table caption,
  consider assigning it.
      - If multiple images or tables seem relevant, choose the single best fit.

Figure 15: Prompt for Mapper.

```
   - If none of the images or tables are relevant, or if none are provided, do not assign anything for that
section.
   3. Produce a single JSON object. Each key is the exact name of a top-level section (e.g.,
"Introduction", "Methods", "Results"), and the value is an object with:
     • "image": image_id or "table": table_id
     • "reason": short explanation describing why the image/table is assigned
   4. If no assignment is made for a section, exclude that section from the JSON.
   6. Ensure your final response strictly follows JSON syntax with no extra commentary.
   7. Keep the original hierarchy (sections → subsections).
   8. Use imageN / reason(N-1) and tableN / reasonTN naming as described.
   9. No image/table reuse limits across subsections, but do not repeat an item twice
     inside the same subsection.

 Example output format if two sections are assigned:
 {
  "sections": [
    {
      "title": "Motivation And Background",
      "subsections": [
        {
          "title": "Challenges in Scientific Video Reconstruction",
          "image1": 1,
          "reason": "Image 1 illustrates sparse sampling and spatiotemporal gaps discussed in this
subsection.",
          "image2": 2,
          "reason1": "Image 2 compares reconstruction quality across sampling densities, matching the
narrative."
        },
        {
          "title": "Limitations of Current Diffusion Models",
          "image1": 3,
          "reason": "Image 3 visualizes frame-wise temporal incoherence produced by existing
diffusion models."
        }
      ]
    },
    {
      "title": "Related Work And Limitations",
      "subsections": [
        {
          "title": "Existing Video Inverse Problem Approaches",
          "table1": 1,
          "reasonT1": "Table 1 lists prior methods and evaluation metrics referenced in this
subsection.",
          "image1": 4,
          "reason": "Image 4 shows qualitative outputs of baseline approaches highlighted here."
        },
        {
          "title": "Plug-and-Play Diffusion Priors",
          "image1": 5,
```

Figure 16: Prompt for Mapper.

```
  "reason": "Image 5 presents an overview diagram of the PnPDP framework emphasized in this
subsection."
        }
      ]
    }
  ]
}

jinja_args:
  - json_content
  - image_information
  - table_information
```

Figure 17: Prompt for Mapper.

# E  USE OF LARGE LANGUAGE MODELS

In accordance with ICLR guidelines, we disclose that Large Language Models (LLMs) were used during the preparation of this manuscript. Their involvement was strictly limited to language and presentation support, including proofreading, grammar correction, and enhancing sentence clarity and readability. The LLMs played no role in the scientific aspects of this work: they did not contribute to the research conception, methodological design, experimental analysis, or the generation of results and conclusions. All substantive ideas, findings, and intellectual contributions are entirely those of the authors.

