# OpenReview forum: "Paper2Slide: A Multi-Agent Framework for Automatic Scientific Slide Generation"
_ICLR.cc/2026/Conference — ICLR 2026 Conference Withdrawn Submission_

### Official Review · Reviewer_zFVu · 2025-10-30

**Soundness:** 2
**Presentation:** 2
**Contribution:** 2
**Rating:** 2
**Confidence:** 4

**Summary:**

The paper introduces SlideGen, a multi-agent framework for automatically generating academic presentation slides from research papers. The framework models the slide-design workflow using six specialized agents—Outliner, Mapper, Formulizer, Arranger, Speaker, and Refiner—each instantiated from the same LLM but prompted with distinct role instructions. These agents collaborate in sequence to transform a paper’s textual and visual content into coherent, well-organized slides.

**Strengths:**

1. Proposes SlideGen, a modular and interpretable LLM-based multi-agent pipeline for text-to-slide generation.
2. Introduces the Paper2Slide benchmark, a paired dataset of research papers and corresponding slides, enabling systematic evaluation of text-to-slide generation methods.

**Weaknesses:**

1.  Appendix B states that “ChatGPT-5o” is used to generate questions, but this model is not publicly standardized; its source and nature are unclear. The paper also does not describe any verification or quality-control procedure for the questions, undermining the objectivity of the evaluation. The PPT style in the Paper2Slide benchmark that SlideQA relies on is highly uniform, which introduces potential limitations (style homogeneity).
2.  In Table 2, non-best results are incorrectly bolded, and the bolding logic is inconsistent. Reported gains are small and lack statistical significance analysis. The table title uses SlideQuiz while the main text uses SlideQA, which is inconsistent. Tables 2 and 3 also lack comparisons to PPTAgent-5 and PosterAgent-5, making the experiments incomplete.
3.  Although the paper discusses possible reasons why GPT-5 underperforms GPT-4o, the backbone comparisons are limited and do not sufficiently demonstrate stability and generalization across models. The human evaluation uses only one person, which further weakens persuasiveness.
4.  The manuscript’s layout and figure/table organization are unclear, leading to choppy logic and poor readability; the paper appears under-polished. Appendix B.2.1 still uses “poster/image,” inconsistent with the main text’s “slide/deck,” suggesting residual terminology from Paper2Poster and blurring the task definition. The paper claims six readers, yet Appendix B.2.1 sets M=3, creating confusion about averaging and reproducibility. Table 3 states the density weights default to λ1=λ2=0.5, while Appendix B fixes λ1=0.6, λ2=0.4, which is inconsistent.
5. Color palettes, fonts, and layouts are highly similar, lacking diversity and professional visual standards; the target style is visibly suboptimal (cf. Figure 3b, GPT-5 HTML).

**Questions:**

1.  You use (λ1,λ2) in Geometry-Aware Density and a token-based length-penalty strength γ in LPA (calibrated via α). To avoid outcomes being driven by parameter choices, please provide a unified robustness report: systematically scan λ1 ⁣: ⁣λ2(from OM-biased to FR-biased) and analyze sensitivity for γ and the length measure LLL, including the γ=0 no-penalty baseline, and state whether the main conclusions remain unchanged.
2. Did you evaluate the Speaker at any point? If not, please explain. The paper also lacks runnable per-module ablations/replacements, error-propagation analysis, or compute-budget curves, making it impossible to distinguish each agent’s necessity and marginal contribution. Please add corresponding evidence; if ablations are infeasible due to system integrity, provide adequate explanations and alternative analyses (e.g., substitute/degraded configurations).
3. Because the question generator and some evaluated models (or same-family models) may coincide, there could be evaluation–evaluator coupling. Please add a brief cross-evaluation (varying the question source and judging the model) plus a small human check to support objectivity and independence.
4. The system appears to produce a largely uniform slide style, which may limit acceptance in real-world scenarios. Do you plan to support multiple style templates or introduce adaptive style generation to meet the needs across users and domains?

---

### Official Review · Reviewer_LRGB · 2025-10-31

**Soundness:** 3
**Presentation:** 3
**Contribution:** 3
**Rating:** 4
**Confidence:** 3

**Summary:**

The paper introduces SlideGen, a visual-in-the-loop agentic pipeline designed to transform complete scientific papers into structured, readable, and well-designed editable slides. To more effectively evaluate the quality of the generated slides, the authors further release the Paper2Slide Benchmark, which consists of paper–slide pairs and provides automated evaluation protocols. On this benchmark, SlideGen achieves strong performance across all evaluation metrics and surpasses various competing methods, demonstrating high-quality slide-generation capabilities.

**Strengths:**

- This paper is well-organized, clearly written, and provides a detailed description of the proposed and used methods.

- The Paper2Slide Benchmark offers a comprehensive evaluation protocol, incorporating multiple metrics that jointly assess narrative quality, factual coverage, visual readability, and quiz-style comprehension, which enables a more holistic evaluation of slide-generation performance.

- On the Paper2Slide Benchmark, the proposed model achieves quantitatively superior results across most experiments, demonstrating its strong capability in generating informative, coherent, and visually appealing slides.

**Weaknesses:**

- The process involving the Arranger, Refiner, Mapper appears somewhat counterfactual, only if the content comprising both textual and visual information is known, it could be ableb to select the most suitable slide templates.

- When the formulas are detected and extracted and their corresponding slides are identified, it remains unclear how the system determines the appropriate placement of these formulas within the slides.

- The Paper2Slide benchmark consists of 200 paper–slide pairs; however, the role of these paired slides is not clearly explained. The proposed method neither utilizes these slides nor includes them in the evaluation process. Given that the corresponding slides are considered high-quality, it would be valuable to compare the model’s outputs against them using metrics such as SlideQA evaluations, geometry-aware density, and other evaluation criteria mentioned in the paper.

- SlideGen was evaluated solely on the Paper2Slide benchmark, which contains only 200 papers. To better assess the generalization capability of the model, it is recommended to further evaluate its performance on other datasets, such as Zenodo10K [1].

[1] Zheng H, Guan X, Kong H, et al. Pptagent: Generating and evaluating presentations beyond text-to-slides[J]. arXiv preprint arXiv:2501.03936, 2025.

**Questions:**

See the Weaknesses

---

### Official Review · Reviewer_HTme · 2025-10-31

**Soundness:** 2
**Presentation:** 2
**Contribution:** 2
**Rating:** 2
**Confidence:** 4

**Summary:**

The paper introduces SlideGen, a multi-agent framework for generating academic presentation slides from a scientific paper, beyond treating it as a text summarization task. SlideGen employs six specialized agents to handle tasks such as content outlining, figure and formula mapping, layout selection, speaker note generation, and slide refinement. To evaluate the quality of generated slides, the authors propose a small benchmark (Paper2Slide) spanning 3 AI conferences over 3-4 editions, as well as VLM-based metrics covering content, design, and coherence; a Q&A-based covering communication effectiveness; and another automated, geometric density-based metric for visual balance. Experimental results mostly favor SlideGen's agent-based design, but are limited by the benchmark's small scale and low diversity, and lack of proper human-in-the-loop validation.

**Strengths:**

1. The slide generation task has received increasing attention, and the goal of approaching it from other perspectives beyond plain text summarization is relevant.

2. Introducing VLM-independent measures such as Geometry-Aware Density is potentially interesting, provided that they demonstrably capture some aspect of the results that humans are shown to value.

3. The multi-agent framework is easy to understand, and Fig. 1 is a particularly clear visual summary.

**Weaknesses:**

1. Evaluation is entirely automatic through a protocol proposed by the authors, with no correlation with human preferences.* Therefore, it is unclear if and to what extent the proposed protocol can serve as a realistic proxy to humans. Authors can refer to the already cited PPTAgent [1], Table 5 and Appendix B, as an example.

2. The Paper2Slide Benchmark also proposed by the authors is very narrow, as it only covers three AI conferences---that is, a single domain with its own specific biases. For contrast, alternative datasets such as in Zenodo10k [1] or AutoPresent [2] cover a greater diversity of domains, making subsequent findings more relevant and sound.

3. Overall, the paper presentation should improve before publication. For example: i) Lines 90-99 appear out of place, closer to what would be expected in a discussion section than in the contribution highlights; ii) More proofreading throughout would be beneficial, such as the typo on line 159 or the missing appendix on line 476; iii) A line plot in Fig. 5 doesn't make sense---a bar plot would have been more appropriate.

*Lines 472-478 allude to a single PhD student reviewing a very small sample, but the appendix is missing and no agreements can be found. (Still, a single human would not allow for inter-annotator agreement to be reported.)

[1] Zheng, Hao, Xinyan Guan, Hao Kong, Jia Zheng, Weixiang Zhou, Hongyu Lin, Yaojie Lu, Ben He, Xianpei Han, and Le Sun. "PPTAgent: Generating and evaluating presentations beyond text-to-slides." arXiv preprint arXiv:2501.03936 (2025).

[2] Ge, Jiaxin, Zora Zhiruo Wang, Xuhui Zhou, Yi-Hao Peng, Sanjay Subramanian, Qinyue Tan, Maarten Sap et al. "Autopresent: Designing structured visuals from scratch." In Proceedings of the Computer Vision and Pattern Recognition Conference, pp. 2902-2911. 2025.

**Questions:**

Could the authors please clarify the human-in-the-loop validation with a PhD student as alluded in lines 472-478? The appendix is missing and I couldn't find the details.

---

### Official Review · Reviewer_nZuG · 2025-11-01

**Soundness:** 3
**Presentation:** 2
**Contribution:** 3
**Rating:** 4
**Confidence:** 3

**Summary:**

This paper presents SlideGen, a modular, multi-agent framework for automatically generating high-quality scientific presentation slides from full research papers. Unlike prior work that treats slide generation as a pure text summarization task, SlideGen explicitly models the visual-textual interplay inherent in effective slide design. The system decomposes the complex end-to-end task into six specialized roles (Outliner, Arranger, Refiner, Mapper, Formulizer and Speaker), each implemented as a vision-language model (VLM) agent. In addition to the framework, the authors contribute Paper2Slide, a new benchmark of 200 high-quality paper-slide pairs from top AI venues and propose a comprehensive evaluation protocol comprising four complementary metrics. Experiments show that SlideGen significantly outperforms strong baselines across almost all metrics, demonstrating high-quality slide-making capability.

**Strengths:**

- This paper correctly identifies that prior work may ignore layout, visual alignment, and narrative flow during slide generation. By reframing it as a multimodal planning and design problem, the work opens a more realistic and useful direction.

- The six-agent modular pipeline is thoughtfully architected, with clear separation of concerns. This design enables interpretability, debuggability, and iterative improvement—key for real-world deployment.

- The release of Paper2Slide fills a void in the community. More importantly, the proposed evaluation suite is innovative and holistic.

**Weaknesses:**

- The refinement process involves slide merging and template adjustment, which may seem intuitive initially but can lead to errors upon closer inspection. For instance, consecutive slides from different sections might be unintentionally merged, or in some cases, slides that contain only text without visual content may still be acceptable for communication. Therefore, it is crucial to analyze the frequency of such occurrences and evaluate whether these processes truly improve the final slide generation.

- Human evaluation is currently limited to one PhD student reviewing five papers. A larger-scale user study would help further validate the real-world utility of the approach. Additionally, there are referring errors in the “Human Evaluation” section, where Table 5 and Figure 6 are incorrectly referenced.

- The experiments focus solely on comparing different automated methods and do not include a direct comparison with human-created slides (e.g., slides generated by the original authors of the same paper versus those produced by SlideGen). This omission leaves a crucial question unanswered: Does SlideGen genuinely achieve “human-level” quality, or does it merely outperform other automated baselines? Despite this, the authors consistently conclude that SlideGen advances automated slide generation toward human-quality results.

**Questions:**

See the Weaknesses

---

### Note · Authors · 2025-11-12

I have read and agree with the venue's withdrawal policy on behalf of myself and my co-authors.